# The Autism Spectrum Disorder-Associated Bacterial Metabolite *p*-Cresol Derails the Neuroimmune Response of Microglial Cells Partially via Reduction of ADAM17 and ADAM10

**DOI:** 10.3390/ijms231911013

**Published:** 2022-09-20

**Authors:** Yuanpeng Zheng, Naika Z. Prince, Lucia N. Peralta Marzal, Sabbir Ahmed, Johan Garssen, Paula Perez Pardo, Aletta D. Kraneveld

**Affiliations:** 1Division of Pharmacology, Utrecht Institute for Pharmaceutical Sciences, Faculty of Science, Utrecht University, 3584 CG Utrecht, The Netherlands; 2Global Centre of Excellence Immunology, Danone-Nutricia Research, 3584 CT Utrecht, The Netherlands

**Keywords:** autism spectrum disorder, valproic acid, *p-*cresol, *p-*cresyl sulfate, ADAM10, ADAM17, neuroimmune response

## Abstract

The bacterial metabolite 4-methylphenol (*para*-cresol or *p*-cresol) and its derivative *p*-cresyl sulfate (*p*CS) are elevated in the urine and feces of children with autism spectrum disorder (ASD). It has been shown that *p*-cresol administration induces social behavior deficits and repetitive behavior in mice. However, the mechanisms of *p*-cresol, specifically its metabolite *p*CS that can reach the brain, in ASD remain to be investigated. The *p*CS has been shown to inhibit LPS-stimulated inflammatory response. A Disintegrin And Metalloprotease 10 (ADAM10) and A Disintegrin And Metalloprotease 17 (ADAM17) are thought to regulate microglial immune response by cleaving membrane-bound proteins. In the present study, a neuroinflammation model of LPS-activated BV2 microglia has been used to unveil the potential molecular mechanism of *p*CS in ASD pathogenesis. In microglial cells *p*CS treatment decreases the expression or maturation of ADAM10 and ADAM17. In addition, *p*CS treatment attenuates TNF-α and IL-6 releases as well as phagocytosis activity of microglia. In in vitro ADAM10/17 inhibition experiments, either ADAM10 or ADAM17 inhibition reduces constitutive and LPS-activated release of TNF-α, TNFR-1 and IL-6R by microglial cells, while it increases constitutive and LPS-activated microglial phagocytotic activity. The in vivo results further confirm the involvement of ADAM10 and ADAM17 in ASD pathogenesis. In in utero VPA-exposed male mice, elevated concentration in serum of *p*-cresol-associated metabolites *p*CS and *p*-cresyl glucuronide (*p*CG) is associated with a VPA-induced increased ADAM10 maturation, and a decreased ADAM17 maturation that is related with attenuated levels of soluble TNF-α and TGF-β1 in the mice brain. Overall, the present study demonstrates a partial role of ADAM10 and ADAM17 in the derailed innate immune response of microglial cells associated with *p*CS-induced ASD pathogenesis.

## 1. Introduction

Autism spectrum disorder (ASD) is a neurodevelopmental disorder characterized by deficits in social communication and interaction, impairments in memory and learning activity, as well as the presence of repetitive behaviors that can have substantial impact on the quality of life [1]. Besides the behavioral and cognitive deficits, autistic individuals often suffer from gastrointestinal problems associated with gut microbiota dysbiosis [2,3]. In recent years, reports point to a possible role of the gut–brain axis in ASD. Recently a thorough epidemiological review showed that the current worldwide ASD prevalence is 1%, which makes ASD one of the most frequently occurring neurodevelopmental disorders in childhood [4]. Additionally, ASD is diagnosed more in males than females with an approximate gender-bias of 4.1:1 [4,5,6]. At present, due to lacking clinically sensitive biomarkers, ASD diagnosis relies on behavioral evaluations and there are no effective treatments targeting the ASD core symptoms. Therefore, identifying therapeutical molecular targets for treatment has become increasingly urgent [7,8,9]. Although the pathophysiology of ASD remains elusive, ASD is known to be caused by an interplay between genetic and environmental factors. A wide range of susceptibility genes have been identified accounting for 10–20% of ASD cases [10,11]. Possible prenatal and postnatal environmental factors associated with an enhanced risk of ASD include maternal exposure to toxins or certain medications (including antibiotics, anti-epileptic medicine and selective serotonin reuptake inhibitors), infections and specific epigenetic factors [11,12,13]. Prenatal exposure to the anti-epileptic medicine valproic acid (VPA) at gestational day 10–13.5 in mice can induce substantial and permanent core symptoms of human ASD, including social impairments, repetitive behavior, and cognitive rigidity, as well as an intestinal phenotype [14,15]. The in utero VPA-induced mice model is a widely recognized ASD model [14,15,16,17]. We and others have reported that in utero exposure to VPA during pregnancy present with social impairments and repetitive behavior, specifically in the male offspring, and reflecting the sex-difference of ASD prevalence in humans [14,18].

In the brain dysregulated immunomodulation of microglia leads to immune abnormalities and synapse dysfunction in the context of ASD pathogenesis [19]. Microglia is an important resident in immune cells in the brain, and regulates inflammatory responses and synaptic pruning, which orchestrate brain development, connectivity and homeostasis [19,20,21]. ASD patients show activated microglia in multiple brain regions [22,23]. Enhanced microglia density also occurs in brain regions of in utero VPA-induced murine models [24,25]. In recent years, it has been suggested that gut microbiota affect microglial maturation and function through producing bacterial metabolites [19,26,27]. The 4-methylphenol (*para*-cresol or *p*-cresol) is a known uremic toxin and a metabolite produced by several bacterial families of the gut microbiota, including Clostridiaceae, Lachnospiraceae and Ruminococcaceae [28,29,30]. Once produced by bacterial fermentation of tyrosine and phenylalanine in the intestine, 95% of *p*-cresol is metabolized by the host into 4-methylphenyl sulfate (*p*-cresyl sulfate, *p*CS) via O-sulfonation, a process that occurs primarily in the liver, and to a smaller extent in colonic epithelial cells [31,32,33]. The bacterial *p*-cresol and its derivative *p*CS are elevated in the urine and feces of ASD children [3,34,35]. In addition, *p*CS has been detected in mice brain tissues, suggesting that *p*CS can enter the brain and might further affect microglial function in the brain [36,37]. Furthermore, *p*-cresol and *p*CS have been proposed as potential biomarkers for ASD clinical diagnosis [35,38]. Pascucci et al. have shown that *p*-cresol (intravenously delivered by tail vein injection) exacerbates ASD-like behaviors in BTBR mice [12]. Very recently, Bermudez-Martin et al. have described how *p*-cresol administration in drinking water induces social behavior deficits and repetitive behavior in mice, by remodeling the gut microbiota [39]. However, the exact molecular mechanism of how *p*-cresol and *p*CS contribute to the pathogenesis of ASD still remains elusive.

ADAM10 and ADAM17 are two members of A Disintegrin And Metalloproteases (ADAMs) family that includes a group of enzymes that are able to cleave membrane-bound proteins. ADAM10 and ADAM17 cleave important protein substrates regulating neuronal networks and immune responses, respectively, such as synaptic molecules Neural Glial-Related Cell Adhesion Molecules (NrCAM) and neuro-inflammatory cytokine Tumor Necrosis Factor α (TNF-α), which are involved or derailed in ASD [40]. ADAM10 plays a key role in the modulation of the molecular mechanisms responsible for dendritic spine formation, maturation and stabilization, and in the regulation of the molecular organization of the glutamatergic synapse [41,42]. Moreover, ADAM10 maturation is increased in a mouse model of Fragile X syndrome (FXS) that is used as a murine ASD model [42]. It is feasible to conclude that the upregulation of ADAM10, resulting in enhanced cleavage of synaptic substrates, might be important in the synaptopathies of ASD. The involvement of ADAM10 in ASD needs to be further studied. In addition to cleaving TNF-α to trigger microglial immune response, ADAM17 regulates microglial phagocytosis capacity by cleaving Triggering receptor expressed in myeloid cells (TREM2) [43,44]. Ray et al. have shown a significantly elevated soluble ADAM17 level in brain tissue of patients with ASD [45], but its role in ASD pathogenesis still remains to be investigated.

Several in vitro studies have shown that *p*-cresol and *p*CS treatment decrease the LPS-induced inflammation and suppresses LPS-activated immune response in murine macrophages [46,47,48], but it is unknown whether *p*CS has similar effects on microglial functions in the brain. The aim of the present study is to investigate the connection between *p*CS, and ADAM10 and ADAM17, in the derailed immune response of microglial cells associated with ASD pathogenesis.

The present study investigated the direct effects of *p*CS on innate immune response and phagocytosis activity of microglial cells, in the absence and presence of LPS stimulation to examine the connection between *p*CS and ADAM10/ADAM17 in the derailed immune response of microglial cells in vitro. In order to demonstrate whether ADAM10 and ADAM17 are affected in in utero VPA-exposed male mice, the expression and maturation of ADAM10 and ADAM17, as well as their downstream neuro-inflammatory cytokines soluble TNF-α and TGF-β1 in the mice brain tissues, were measured (Appendix A and results).

## 2. Results

### 2.1. Effect of pCS on the Expression of ADAM10 and ADAM17 of Microglial Cells Constitutively and during LPS-Induced Inflammation

The *p*-cresol host metabolites *p*CS and *p*CG levels were significantly increased in in utero VPA-exposed male mice, when compared to control mice (Appendix A). Decreased imADAM10, increased mADAM10 and ADAM10 maturation efficiency were observed in the hippocampus and other brain regions of in utero VPA-exposed male mice compared to control mice (Appendix A). In addition, in utero exposure to VPA significantly decreased both mADAM17 content and ADAM17 maturation efficiency in the hippocampus, compared to control mice (Appendix A). ADAM17, also called TNF-α Converting Enzyme (TACE) [49], cleaves membrane bound TNF-α to soluble TNF-α (sTNF-α) and is an important regulator of TGF-β1 [50,51]. The VPA-reduced ADAM17 expression was associated with decreased expression of sTNF-α and TGF-β1 levels in the hippocampus (Appendix A). These results show that enhanced systemic *p*CS is connected with dysregulated ADAM10/ADAM17 and ADAM17-associated cytokines in the brain, and is involved with the pathogenesis of in utero VPA-induced ASD.

Because *p*CS can enter the brain [36,37], we therefore examined the connection between *p*CS and ADAM10/ADAM17 in microglial cells. Using two cell viability assays (MTT and LDH), it was demonstrated that 24 h exposure of BV2 cells up to a concentration of 500 μM *p*CS did not affect the viability of BV2 cells (Appendix A).

Exposure of BV2 cells to LPS showed a decrease in mADAM10, but showed no significant effect on imADAM10 and ADAM10 maturation efficiency (mADAM10/imADAM10) when compared to vehicle control (Figure 1A–H). The expression of imADAM10 and mADAM10 was barely affected by exposure to LPS-stimulated BV2, with *p*CS low (0.1, 0.5, 1 and 5 μM, Figure 1A–C) and high (5, 10, 50 and 150 μM, Figure 1E–G) concentrations compared to LPS control; this was also reflected by no significant change in ADAM10 maturation efficiency (Figure 1D,H). In contrast, *p*CS alone at the low concentrations reduced mADAM10 without affecting imADAM10 expression and ADAM10 maturation efficiency (Figure 1A–D). Furthermore, *p*CS alone at the high concentrations reduced the expression of imADAM10 and mADAM10 without affecting ADAM10 maturation efficiency (Figure 1E–H).

The 24 h incubation with LPS induced upregulation of ADAM17 expression in BV2 cells (Figure 1J,L). Co-incubation of LPS-stimulated BV2 cells with *p*CS at high concentrations (5, 10, 50 and 150 μM) demonstrated a reduced expression of ADAM17 (Figure 1L). Moreover, these concentrations of *p*CS reduced the constitutive ADAM17 expression in BV2 cells (Figure 1J,L).

These results indicate that *p*CS might negatively affect the innate immune response of BV2 microglia through inhibiting ADAM17, and possibly through ADAM10. Given that ADAM10 and ADAM17 cleave TNF-α and other cytokine-related substrates, such as IL-6 receptor (IL-6R) and Tumor necrosis factor receptor 1 (TNFR-1) [52,53,54], we further investigated the potential role of ADAM10 and ADAM17 in innate immune and phagocytotic responses of BV2 microglial cells constitutively and after LPS stimulation.

### 2.2. ADAM10 and ADAM17 Cleave TNF-α, TNFR-1 and IL-6R from BV2 Microglia

The role of ADAM10 and ADAM17 is not known in the release of TNF-α and IL-6 as wells as the cleavages of TNFR-1 and IL-6R from BV2 microglial cells during an innate immune response. To this end, we employed the effects of GI254023X, a selective and potent inhibitor of ADAM10 activity [55,56], and TAPI-1, a compound preferential inhibiting ADAM17, followed by other metalloproteinases [57,58] on constitutive and LPS-activated BV2 cells. A 24 h incubation with LPS resulted in a significant release of TNF-α, IL-6 and TNFR-1 compared to vehicle treated BV2 cells, whereas LPS did not affect sIL-6R levels (Figure 2A–D). Both ADAM10 inhibitor, GI254023X (5 μM), and ADAM17 inhibitor, TAPI-1 (25 μM) significantly decreased the constitutive and LPS-induced release of TNF-α and soluble TNFR-1 (sTNFR-1), indicating either ADAM10 or ADAM17 (and possibly other metalloproteases) regulate the cleavages of TNF-α and TNFR-1 in BV2 microglia (Figure 2A,B). GI254023X did not affect the LPS-induced release of IL-6 from BV2 cells, and TAPI-1 significantly enhanced the release of IL-6 (Figure 2C). In contrast, GI254023X and TAPI-1 did inhibit the release of soluble IL-6R (sIL-6R), indicating ADAM10 and ADAM17 control the cleavage of IL-6R (Figure 2D).

In the cell lysates, we further investigated the effect of two inhibitors on the expression of ADAM10 or ADAM17 in BV2 microglial cells constitutively and under inflammation. Figure 2F shows imADAM10 was not affected by all treatments. LPS induced a reduced ADAM10 maturation efficiency (Figure 2H) confirming the findings presented in Figure 1. GI254023X and TAPI-1 significantly decreased constitutively and under inflammation mADAM10 resulting in reduced ADAM10 maturation efficiency (Figure 2G,H). Figure 2J demonstrates that LPS stimulation significantly increased ADAM17 expression compared to vehicle treated BV2 cells, as also demonstrated in Figure 1. Both inhibitors GI254023X and TAPI-1 did not decrease ADAM17 expression of BV2 cells constitutively or under inflammation (Figure 2J). Furthermore, GI254023X on its own significantly increased ADAM17 expression compared to vehicle treated BV2 cells (Figure 2J).

### 2.3. The Effect of ADAM10 and ADAM17 Inhibition on Constitutive and LPS-Induced Phagocytosis by BV2 Cells

As shown in Figure 3, 24 h LPS stimulation significantly increased phagocytosis by BV2 cells. GI254023X significantly potentiated LPS-induced phagocytosis compared to LPS (Figure 3A). In addition, TAPI-1 also significantly potentiated LPS-induced phagocytotic activity (Figure 3B). Furthermore, we demonstrated a small but significantly increased constitutive microglial phagocytosis by GI254023X compared to vehicle control (Figure 3A). However, TAPI-1 treatment seemed not to significantly increase constitutive microglial phagocytosis (Figure 3B).

### 2.4. Effect of pCS Exposure on Innate Immune Response of Microglial Cells

We investigated the effect of *p*CS (0.1, 0,5, 1, 5, 10, 50 and 150 μM) on constitutive and LPS-induced TNF-α and IL-6 releases by BV2 microglia. Figure 4A,B demonstrated that LPS stimulation significantly increased the releases of TNF-α and IL-6 from BV2 microglia. Furthermore, Figure 4A showed that *p*CS from 1 μM to 50 μM significantly decreased LPS-induced TNF-α production by BV2 cells. In parallel, *p*CS from 5 μM to 50 μM significantly decreased constitutive TNF-α concentration compared to vehicle control. In addition, 5 to 150 μM *p*CS decreased LPS-induced IL-6 release by BV2 cells (Figure 4B). Meanwhile, *p*CS at concentrations of 1, 5, 10 and 50 μM decreased the basal IL-6 production compared to vehicle control.

### 2.5. Effect of pCS on Phagocytosis Response of Microglial Cells

24 h LPS stimulation of BV2 cells significantly increased phagocytosis by BV2 cells (Figure 5). Co-incubation of BV-2 cells with *p*CS and LPS did not result in a significantly different phagocytotic response compared to LPS control, demonstrating that *p*CS cannot affect the LPS-induced phagocytotic response (Figure 5). In contrast, Figure 5B shows that *p*CS (10, 50 and 150 μM) significantly decreased constitutive phagocytotic activity of BV2 cells compared to vehicle control.

## 3. Discussion

The present study investigated the effects of *p*CS on innate immune response and phagocytosis activity of microglial cells, which is partially linked to ADAM10 and 17 functions that are dysregulated in the brain tissues of a murine model of ASD. Previous findings showed that *p*CS or *p*-cresol decreases LPS-activated immune response of murine macrophage [46,47,48]. This study showed that *p*CS inhibited constitutive and LPS-induced releases of TNF-α and IL-6 from microglial cells. To date, this is the very first report showing that the *p*CS-induced derailed immune response of microglial cells is associated with reduced expression or maturation of ADAM17, and to a lesser extent of ADAM10. To further explore the role of ADAM10 and ADAM17 in immune responses of BV2 microglial cells, the effects of respective inhibitors GI254023X and TAPI-1 on LPS-stimulated microglial cells has been studied as well. ADAM10 and ADAM17 collectively cleaved TNF-α, TNFR-1 and IL-6R of constitutive and LPS-activated microglial cells. This finding is in line with previous findings that ADAM17 has been confirmed as the primary sheddase cleaving TNF-α, with concurrent involvement of ADAM10 in mouse macrophages [52]. The sTNF-α forms homotrimers to recognize TNFR to regulate immune response [59,60]. TNFR has also been identified as a substrate for ADAM10 [54], and the current study shows that ADAM17 cleaved TNFR-1 as well of microglial cells. IL-6 binds to IL-6R to trigger an immune response, and during this process IL-6 signaling needs the involvement of membrane-bound IL-6R to form the IL-6/IL-6R/gp130 complex [61]. ADAM17 mainly controls these through cleavage of membrane bound IL-6R to soluble IL-6R [53,62], and indeed, both ADAM17 and ADAM10 seemed to play a role in the cleavage of IL-6R from microglial cells. Unexpectedly, the TAPI-1 increased LPS-stimulated IL-6 concentration in BV2 cells, which is consistent with previous findings, showed that TAPI-1 increases LPS-stimulated IL-6 level in monocytes [63]. In the present study, TAPI-1 induces a decrease in sIL-6R release under LPS stimulation. This will result in more membrane-bound IL6-R expression. Therefore, we suggest that the TAPI-1-induced upregulation of IL-6 release by LPS exposed BV2 cells can be explained by the activation of membrane bound IL6-R, resulting in more IL-6 release. Previously, it has been demonstrated that IL-6 can induce IL-6 secretion by epithelial cells [64]. In addition, LPS treatment increased ADAM17 expression of microglial cells, indicating the involvement of ADAM17 in LPS-induced neuroinflammation of microglial cells. In contrast, LPS stimulation triggered a significant decrease in ADAM10 maturation in microglial cells, which is consistent with findings in macrophages [52]. TAPI-1 and GI254023X inhibited ADAM10 activity via attenuating its maturation in constitutive and activated microglial cells, which is in line with recent findings showing that these two ADAM inhibitors reduced ADAM10 maturation in monocytes, and that GI254023X treatment inhibits ADAM10 maturation in vitro and in vivo [65]. GI254023X significantly increased ADAM17 expression in microglial cells, which is in line with previous finding that has shown an overcompensation of ADAM17 in ADAM10 deficiency lymph nodes [66]. In addition, ADAM10 functions as the major TNF-α sheddase in ADAM17-deficient fibroblasts [67]. The decreased mADAM10 triggered by GI254023X observed in the present study might increase ADAM17 expression reversely. As TAPI-1 might also decrease mADAM10, it also might trigger a lower increase in ADAM17 expression at this point. In summary, ADAM10 and ADAM17 collectively cleave TNF-α, TNFR-1 and IL-6R in constitutive and activated microglial cells, indicating that ADAM10 and ADAM17 might regulate immune responses of microglial cells collectively. Regarding TNF-α, the effect of *p*CS on microglial cells is comparable to that of ADAM17 and ADAM10 inhibitors, suggesting that via these ADAMs *p*CS can derail the innate immune response of microglia observed in ASD [46,68].

Besides innate immune responses, microglial cells have phagocytotic capacities as well which are important during brain development through regulation of synaptic pruning. In the current study, *p*CS only inhibited constitutive microglial phagocytosis, without affecting LPS-induced phagocytosis. Similar results have been reported for *p*CS and phagocytosis of macrophages [46]. Impaired synaptic function due to a reduction in microglia is found to be associated with impaired connectivity and ASD-like behaviors [69]. ADAM10 and ADAM17 seemed not to play a role in the *p*CS-induced reduction of phagocytosis, since both ADAM10 and ADAM17 inhibitors enhanced the LPS-induced phagocytosis of microglial cells. ADAM proteases inhibition using the nonspecific ADAM inhibitor GM6001 and increases microglial phagocytosis [44]. Moreover, recently it has been demonstrated in mice that postnatal enhanced phagocytotic activity of microglial cells is associated with ASD-like behavior [70]. These findings pave the way for our findings suggesting both ADAM10 and ADAM17 play a crucial role in regulating microglial phagocytosis, which is not affected by *p*CS.

Ample studies have suggested that the microbiota–gut–immune–brain axis plays an important role in ASD pathogenesis. The supplementary in vivo results in this report further supported the link between bacterial metabolite derived *p*CS and ADAM10/17 in the VPA-induced ASD model in mice. It is reported that various species within the Firmicutes family of Clostridiaceae are involved in *p*-cresol production from tyrosine and phenylalanine via fermentation [71,72,73,74]. Elevated levels of Clostridiaceae species in fecal samples from ASD patients have been reported [75,76], and several studies have shown that *p*-cresol metabolites*,*
*p*CS or *p*CG, are elevated in urine and feces of autistic children associated with altered gut microbiota composition [34,35,77]. *P*-cresol alters dopamine metabolism, enhances glutamine, and decreases γ-aminobutyric acid in autistic children [12,78,79]. It has been shown that ADAM10 activity plays an important role in regulating glutamatergic synapses [41,80]. Excessive ADAM10 activity hampers spine maturation and impairs synaptic plasticity through cleaving more APP into sAPPα to promote Glutamate Receptor 5 Signaling in a mouse model of ASD [42]. Therefore, the increased mADAM10 might relate to the *p*-cresol-induced decreased glutamine in in utero VPA-exposed male mice. In addition, glutamate activates N-methyl-D-aspartate (NMDA) receptors of microglia to trigger morphological activation and release of inflammatory mediators [81,82,83]; the decreased ADAM17 activity might associate with *p-cresol*-induced decrease in glutamine or increase in γ-aminobutyric acid, and might participate in these inflammatory processes in in utero VPA-exposed male mice. These remain to be investigated. Given the VPA-induced elevated *p*CS level in the serum of male mice and the direct effects of *p*CS on ADAM10/ADAM17 in BV2 microglial cells, this points to an underlying mechanism of the intestinal bacterial metabolite *p*-cresol in the pathogenesis of ASD to be associated with ADAM10/ADAM17 in the brain.

ADAM10 is the main α-secretase shedding APP to generate soluble amyloid precursor protein-α (sAPPα) [84,85]. The sAPPα is upregulated during spine formation and plays a pivotal role in synaptogenesis leading to increased spine density [86,87]. Alteration of spine number and morphology is believed to underlie many neurological disorders including ASD [88]. Moreover, increased mature ADAM10 has been shown to dysregulate APP cleavage to induce synaptic dysfunction in mice, deficient in the Fragile X mental retardation protein (FMRP) that leads to Fragile X syndrome and ASD-like behaviors [42]; this supports the present findings that ADAM10 maturation was increased in in utero VPA-exposed male mice brain. The increased mADAM10 expression can trigger synaptic dysfunction in the brain through cleaving additional synaptic substrates, including NrCAM, Neuroligins, Neurexins and Protocadherins [84,89,90]. To further unravel the potential role of ADAM10 in ASD-associated changes in synaptic structures is beyond the scope of this study.

Regarding ADAM17, increased levels of soluble ADAM17 α-secretase in the brains of ASD patients has been reported [45]. ADAM17 can be cleaved by other ADAMs to be transformed into soluble ADAM17 [91,92], and this can result in decreased membrane-bound ADAM17 levels in the brain concurrently [45,93,94]. These findings are in line with the decreased mADAM17 expression in the brain of in utero VPA-exposed male mice. ADAM17 is important for the cleavage of membrane-bound cytokines [95]. The VPA-induced decreased hippocampal mADAM17 level was associated with decreased hippocampal TGF-β1 levels, which is consistent with the attenuated TGF-β1 in the circulation of ASD children [96,97]. In contrast to the present findings that soluble TNF-α was reduced in hippocampus of in utero VPA-exposed male mice, an increased expression of TNF-α mRNA in both cerebellum and hippocampus has been reported [98], as well as an increase in TNF-α in the brain cortex of ASD patients [99]. This contrasting result might be attributed to different measurements; in previous studies all TNF-α forms at gene or total protein level were measured, but this present study measured soluble TNF-α levels, excluding the membrane-associated form.

In conclusion, this present study demonstrates that in microglial cells *p*CS attenuated the expression or maturation of ADAM10/17 that control the cleavages of TNF-α, TNFR-1 and IL6R. Indeed, *p*CS inhibited the release of TNF-α by microglial cells. Our in vitro results are confirmed by the decreased mADAM17 and the attenuated downstream cytokines sTNF-α and TGF-β1 in the brain tissue of mice in utero exposure to VPA. In addition, *p*CS inhibited IL-6 release of constitutive and LPS-activated BV2 microglial cells and attenuated phagocytosis capacity of constitutive BV2 microglial cells. The *p*CS-induced decrease of ADAM10 in microglial cells does not reflect the increased ADAM10 maturation in the brain of in utero VPA-exposed male mice. This suggests that ADAM10 maturation might be enhanced in other brain cells, such as neurons, which makes sense regarding the important role of ADAM10 in synapse regulation [84,89]. In addition, mice lacking ADAM10 also shows synaptic dysfunction, altered brain connectivity in the cortex and hippocampus [100,101].

To date, this is the very first study that reports on the possible involvement of *p*CS-induced changes in ADAM10 and ADAM17 in the derailed neuroimmune response in context with ASD, which also sheds light on identifying ADAM10 and ADAM17 as potential targets for ASD treatment. Further studies are needed to examine the direct causal link between *p*-cresol, ADAMs activity and ASD development via the microbiota–gut–brain axis. In addition, targeting intestinal *p*-cresol producing bacteria with microbiome-based therapies might reduce its possible detrimental effect on ADAM10 and ADAM17 in the CNS, and might therefore reduce ASD symptoms.

## 4. Materials and Methods

### 4.1. BV2 Microglial Cell Culture and Treatments

BV2 cells were cultured as described before [102] at 37 °C and 5% CO_2_ in medium (Dulbecco’s modified eagle medium (Gibco, Grand Island, NY, USA), 10% Fetal bovine serum (Gibco, Grand Island, NY, USA) and 1% penicillin/streptomycin (Gibco, Grand Island, NY, USA)). BV2 cells were used to investigate possible effects of *p*CS on cell viability, on ADAM10 and ADAM17 expression, on the inflammatory response as well as on phagocytosis.

### 4.2. BV2 Microglial Cell Viability

To assess the effect of *p*CS on cell viability, 5000 BV2 cells/well were seeded into 96-well plate (3599, Corning, NY, USA). On the following day the BV2 cells were incubated with *p*CS (concentration range: 5–150 μM) for 24 h. After 24 h, 50 μL medium of each well was transferred into a new 96-well plate and the content of lactate dehydrogenase (LDH) in medium was measured by Cytotoxicity Detection KitPLUS (LDH) (4744926001, Merck Life Science N.V. Amsterdam, the Netherlands) according to the manufacturer’s instructions. In the meantime, the medium leftover was discarded and 100 μL DMEM containing 0.5 mg/mL MTT (M2128, Sigma) was added to the cells for 4 h incubation at 37 °C under 5% CO_2_. The DMEM was removed and then 200 μL DMSO was added into each well. Finally, the OD values were measured at wavelength of 570 nm. In all viability experiments, 1 μM and 10 μM Rotenone dissolved in DMSO was employed as positive control [103,104,105].

### 4.3. BV2 Microglial ADAM10 and ADAM17 Expression and Inflammatory Response

To investigate the effect of *p*CS on the expression of ADAM10 and ADAM17, 50,000 BV2 cells/well were seeded in a 12-well plate (3512, Corning, NY, USA). To assess the effect of *p*CS on the release of TNF-α and IL-6, 5000 BV2 cells/well were seeded in a 96-well plate (3599, Corning, NY, USA). The cells were incubated with *p*CS (concentration range: 0.1–150 μM) for 24 h in the presence or absence of 1000 ng/mL LPS stimulation (L3024, Sigma). The medium in the 96-well plate was collected for measuring of TNF-α and IL-6 release by ELISA. The BV2 cells in a 12-well plate were lysed for Western blotting analysis.

For ADAMs inhibition experiments, 50,000 BV2 cells/well were seeded into a 12-well plate overnight and incubated with 5 μM GI254023X [106] (ADAM10 inhibitor SML0789, Sigma) or 25 μM TAPI-1 [107] (ADAM17 inhibitor B4686, Novus Bio-Techne, Abingdon, United Kindom) for 24 h with or without 1000 ng/mL LPS stimulation on the following day. The medium was collected for ELISA measurements and the BV2 cells lysed for Western blotting analysis.

All BV2 cell samples used for Western blotting analysis were lysed on ice for 30 min with RIPA buffer (20188, Sigma) containing 1% TritonX-100 detergent, Proteinase Inhibitor Cocktail (1:200), 5 μM GI254023X and 10 mM 1,10-Phenanthroline.

### 4.4. Phagocytosis Activity Assay

To assess the effect of *p*CS on phagocytosis, 50,000 BV2 cells/well were seeded into a 96-well plate (3599, Corning, NY, USA), immediately followed by incubation with *p*CS (concentration range: 0.1–150 μM) for 24 h in the presence or absence of 1000 ng/mL LPS stimulation. To study the effect of ADAM10 and ADAM17 inhibition on constitutive and LPS-stimulated BV2 cells, BV2 cells were incubated with GI254023X (1 μM or 5 μM) or TAPI-1 (5 μM or 25 μM), respectively, for 24 h with or without 1000 ng/mL LPS. BV2 microglial phagocytotic effect was measured with a Vybrant™ Phagocytosis Assay Kit (V6694, Thermo Scientific, Waltham, MA, USA) according to the manufacturer’s instructions. Briefly, the medium was discarded completely after 24 h incubation, and then incubated with K-12 strain bioParticles for 2 h at 37 °C, followed by Trypan Blue solution incubation for 1 min. Finally, the fluoresce intensity was measured using ~480 nm excitation, ~520 nm emission.

### 4.5. Cytokine and Cytokine Receptor ELISAs

(Soluble) TNF-α, TGF-β1, IL-6, IL-6R and TNFR-1 in the medium of BV2 cells or in the supernatants of brain tissue homogenates (see Appendix A) were quantified using mouse TNF-α ELISA kit (430901, BioLegend, San Diego, CA, USA), human/mouse TGF-β ELISA kit (88-8350-88, Invitrogen, Waltham, MA, USA), mouse IL-6 ELISA kit (431301, BioLegend, USA), mouse IL-6R ELISA Kit (RAB0314, Sigma) and Mouse/Rat TNF R-1 Quantikine ELISA Kit (MRT10, R&D Systems), using manufacturer’s instructions. Due to the strong inhibition of constitutive IL-6 release from BV2 microglial cell by *p*CS (1, 5, 10 and 50 μM) treatment, IL-6 concentrations in the medium after exposure to *p*CS (1, 5,10 and 50 μM) were under the detection limit that was set as ‘0′.

### 4.6. Immunoblotting Brain Tissue and BV2 Microglial Cells

As described previously [89,108], the protein concentrations in BV2 microglial cell supernatants and brain tissue (see Appendix A) were quantified by Pierce™ BCA Protein Assay Kit (23225, Thermo Scientific, Vantaa, Finland). Then the supernatant was denatured at 95 °C for 5 min in 4X Laemmli Sample Buffer (1610747, Bio-Rad, USA) containing 50 mM Dithiothreitol (1610611, Bio-Rad, Hercules, CA, USA). Afterwards, 10 to 30 μg of protein was loaded and separated on 4–15% gradient precast polyacrylamide gels (5671084, Bio-Rad, USA). Next, proteins were transferred to PVDF membranes (1704157, Bio-Rad, USA) with Trans-Blot Turbo Transfer System. The membranes were blocked in 5% skim milk-PBST for 1 h and then incubated with the primary antibody overnight at 4 °C. The membranes were washed with PBST buffer and incubated with anti-rabbit (1:3000, Dako, P0448, USA) or anti-mouse (1:3000, Dako, P0260, USA) secondary antibodies for 1 h at room temperature. After washing, the membranes were exposed to Clarity Western ECL Blotting Substrate (1705060, Bio-Rad Laboratories, Hercules, CA, USA), and were imaged using the ChemiDoc MP Imaging System (Bio-Rad Laboratories, Hercules, CA, USA) to detect bands. Image J software (version 1.52v, National Institutes of Health, Bethesda, USA) was used to quantify the density of the bands. The following antibodies were used: ADAM10 primary antibody (1:1000, ab124695, Abcam, Cambridge, UK); TACE/ADAM17 primary antibody (NBP2-15281, 1:500, Novus Biologicals, used for BV2 microglia samples); ADAM17 primary antibody (AB19027, 1:1000, Sigma, used for brain tissue samples); β-Actin primary antibody (1:3000, MA5-15739, Invitrogen, USA) or Calnexin primary antibody (1:3500, PA5-34754, Invitrogen, USA) as loading control.

### 4.7. Statistics

All data analysis and statistics were performed using GraphPad Prism (version 9.1.1; GraphPad software, La Jolla, CA, USA). For multiple comparisons of the in vitro data, one-way ANOVA was used with Dunnett’s multiple comparison test. The in vivo results were analyzed by two-tailed Student’s test. The results were expressed as mean ± SEM. *p* < 0.05 is considered to be statistically significant.

## Figures and Tables

**Figure 1 ijms-23-11013-f001:**
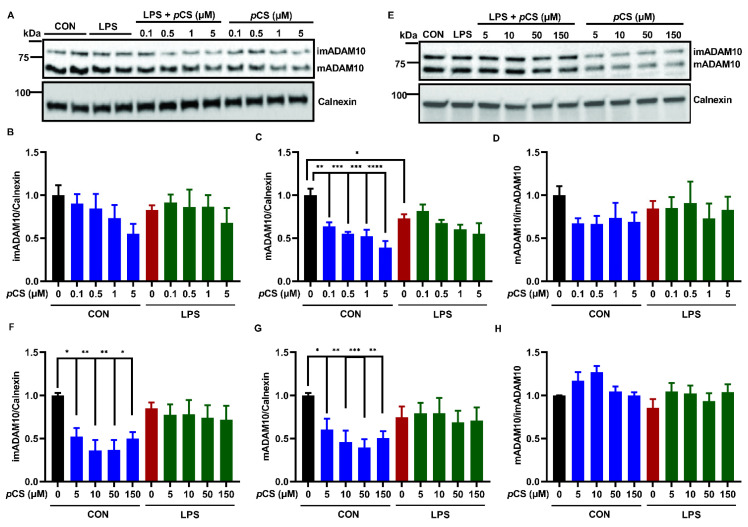
Effect of *p*CS on ADAM10 and ADAM17 expression of constitutive and LPS-activated BV2 microglial cells. BV2 microglia were incubated with *p*CS for 24 h in the absence or presence of 1000 ng/mL LPS. Cell lysate was collected for WB analysis. (**A**) The representative immunoblots of ADAM10 protein expression with low concentration range of *p*CS. (**B**–**D**) The quantification results of immature ADAM10 (imADAM10), and mature ADAM10 (mADAM10) and maturation efficiency (ratio mADAM10/imADAM10) of BV2 cells exposed to a low concentration range of *p*CS (*n* = 4 independent experiments). (**E**) The representative immunoblots of ADAM10 protein expression with a high concentration range of *p*CS. (**F**–**H**) The quantification results of ADAM10 protein expression of BV2 cells exposed to a high concentration range of *p*CS treatment (*n* = 5 independent experiments). (**I**) The representative immunoblots of ADAM17 protein expression with low concentration range of *p*CS. (**J**) The quantification results of ADAM17 protein expression with low concentration range of *p*CS (*n* = 4 independent experiments). (**K**) The representative immunoblots of ADAM17 protein expression with a high concentration range of *p*CS. (**L**) The quantification results of ADAM17 protein expression with a high concentration range of *p*CS (*n* = 6 independent experiments). Calnexin or β-Actin were used as a loading controls. Black: constitutive protein expression set at 1; red: LPS-induced protein expression; blue: effect of *p*CS on constitutive protein expression; green: effect of *p*CS on LPS-induced protein expression. Results were expressed as mean ± SEM. ** p* < 0.05, *** p <* 0.01, **** p* < 0.001*, **** p* < 0.0001.

**Figure 2 ijms-23-11013-f002:**
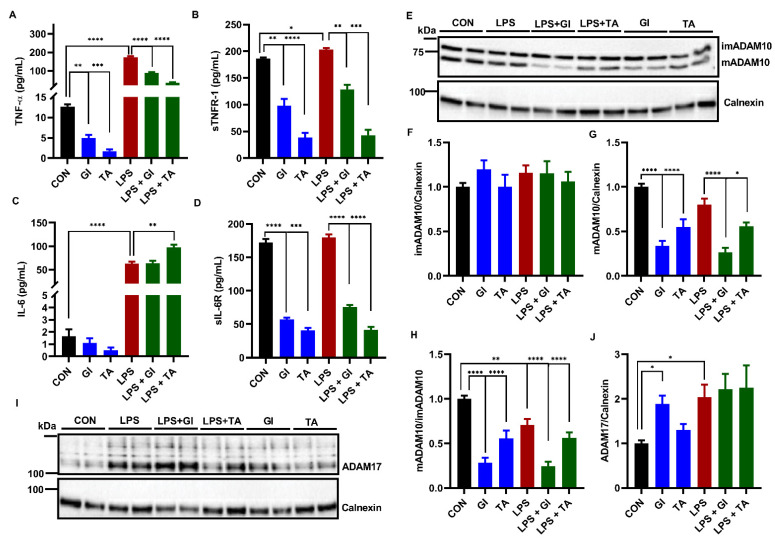
The effect of ADAM10 inhibitor (GI254023X) or ADAM17 inhibitor (TAPI-1) on the release of TNF-α, TNFR-1 and IL-6R by constitutively and under inflammation BV2 microglia. BV2 microglia were incubated with GI254023X (5 μM) or TAPI-1 (25 μM) for 24 h in the absence or presence of 1000 ng/mL LPS. The medium was collected for ELISA measurements and cell lysates were lysed for WB analysis. The concentration of TNF-α (**A**), sTNFR-1 (**B**), IL-6 (**C**) and sIL-6R (**D**) released in medium (*n* = 6 from 6 independent experiments). (**E**) The representative immunoblots of ADAM10 protein expression in BV2 cell lysate. (**F**–**H**) The quantification of ADAM10 expression in BV2 cell lysate (*n* = 10 from 5 independent experiments). (**I**) The representative immunoblots of ADAM17 expression in BV2 cell lysate. (**J**) The quantification results of ADAM17 in BV2 cell lysate (*n* = 8 from 4 independent experiments). Calnexin was used as a loading control. Black: constitutive cytokine release or protein expression set at 1; red: LPS-induced cytokine release or protein expression; blue: effect of ADAMs inhibitor on constitutive cytokine release or protein expression; green: effect of ADAMs inhibitor on LPS-induced cytokine release or protein expression. Results were expressed as mean ± SEM. ** p* < 0.05, *** p* < 0.01 **** p* < 0.001*, **** p* < 0.0001.

**Figure 3 ijms-23-11013-f003:**
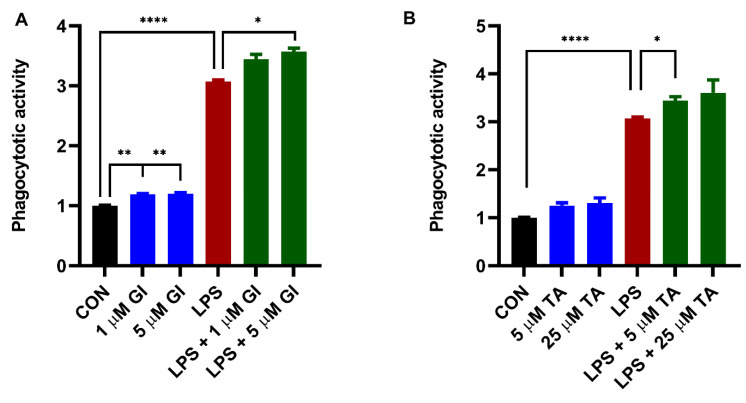
The effect of ADAM10 inhibitor (GI254023X) or ADAM17 inhibitor (TAPI-1) on constitutively and LPS-induced BV2 microglial phagocytosis activity. BV2 microglial cells were incubated with GI254023X (1 and 5 μM) or TAPI-1 (5 and 25 μM) for 24 h in the absence or presence of 1000 ng/mL LPS. (**A**) The effect of GI254023X treatment on BV2 microglial phagocytosis activity. (**B**) The effect of TAPI-1 treatment on BV2 microglial phagocytosis activity. Black: constitutive phagocytes set on 1; red: LPS-induced phagocytosis; blue: effect of ADAMs inhibitor on constitutive phagocytosis; green: effect of ADAMs inhibitor on LPS-induced phagocytosis. Results were expressed as mean ± SEM. *n* = 4 from 4 independent experiments. ** p* < 0.05, *** p* < 0.01, ***** p* < 0.0001.

**Figure 4 ijms-23-11013-f004:**
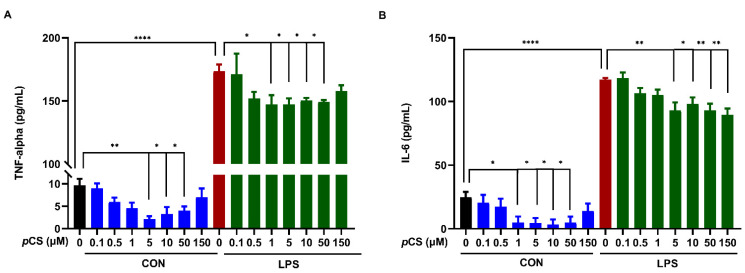
The effect of *p*CS on TNF-α and IL-6 releases by BV2 microglia. The BV2 microglial cells (5.000 per well) were incubated with *p*CS for 24 h with or without LPS (1000 ng/mL). (**A**) The concentration of TNF-α in culture medium: *n* = 3 from 3 independent experiments. (**B**) The concentration of IL-6 in culture medium: *n* = 4 from 4 independent experiments. Black: constitutive cytokine release; red: LPS-induced cytokine release; blue: effect of *p*CS on constitutive cytokine release; green: effect of *p*CS on LPS-induced cytokine release. Results are expressed as mean ± SEM. ** p* < 0.05, *** p* < 0.01*, **** p* < 0.0001.

**Figure 5 ijms-23-11013-f005:**
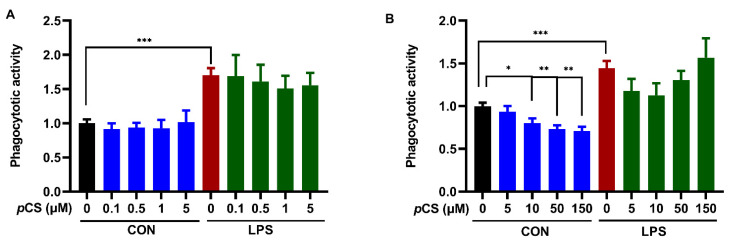
The effect of *p*CS on constitutive and LPS-induced (1000 ng/mL) phagocytosis response of BV2 microglia. (**A**) The effect of *p*CS (0.1, 0.5, 1, 5 μM) on phagocytosis response of BV2 microglia (*n* = 4 independent experiments). (**B**) The effect of *p*CS (5, 10, 50 and 150 μM) on phagocytosis response of BV2 microglia (*n* = 6 independent experiments). Black: constitutive phagocytes set on 1; red: LPS-induced phagocytosis; blue: effect of *p*CS on constitutive phagocytosis; green: effect of *p*CS on LPS-induced phagocytosis. Results were expressed as mean ± SEM. ** p* < 0.05, *** p* < 0.01, **** p <* 0.001.

## Data Availability

The raw data that support the findings of this study will be made available upon reasonable request from the corresponding author.

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
