# Peer review of "The Autism Spectrum Disorder-Associated Bacterial Metabolite p-Cresol Derails the Neuroimmune Response of Microglial Cells Partially via Reduction of ADAM17 and ADAM10"

_ijms, 2022, doi:10.3390/ijms231911013_

Round 1
Reviewer 1 Report
Authors consider the present study of interest for autism because metalloprotease 10 (ADAM10) and metalloprotease 17 (ADAM17) regulate microglial immune response by cleaving membrane-bound proteins (of microglia ?). The relationship between autism and the effects of ADAM10 and ADAM17 should be better emphasized in the introduction in order to allow the reader to appreciate the effects reported in the study. By listing the clinical manifestations of autistics, can you link observed decreases or increases in some chemicals with some of these pathological behaviours?
In autistic children P-cresol and its derivative p-cresyl sulfate (pCS) are elevated in the urine and feces. They reach the brain. In the microglia pCS (and p-cresol ?) decreases ADAM10 and ADAM17 expression. As a consequence, both (and not pCS alone) attenuate TNF-a and IL-6 release as well as phagocytosis activity of microglia. Moreover, (both or) pCS reduce (s) constitutive and LPS-activated release of TNF-a, TNFR-1 and IL-6R by microglia cells while increase (s) constitutive and LPS-activated microglial phagocytic activity.
Please, better clarify in the abstract and introduction.
The study is well documented and supported by compelling results. The study is well documented and supported by compelling results. Thus, the manuscript deserves to be published, although there are repetitions and the Discussion should be improved.
Remove repetitions in the Discission at lines 404- 406, 413-417.
Sentence at line 422-425 belongs to Introduction.
One last concern. It is well known that p-cresol blocks dopamine beta-hydroxylase (C. Southan et al. , Biochim. Biophys. Acta 1037 (2) (1990) 256–258) altering brain dopamine metabolism (Pascucci et al. , Brain Sci 13 (10(4)) (2020)), increasing amount of homovanillic acid, decreasing noradrenaline and adrenaline production, as well as MHPG and vanillylmandelic acid, which were found lower and finally biotransformation of glutamate into γ-aminobutyric acid (GABA). All these neurotransmitters altered neuronal transmission (Gevi. Et al BBA - Molecular Basis of Disease 1866 (2020) 165859).
The correlation between decreased neurotransmitters and autism is very clear. How can you relate p-cresol's decreased neutransmitters to the affects you documented in your study?
Identifying clinically sensitive biomarkers for ADS diagnosis are getting increasingly urgent. Some biochemical biomarkers have been proposed, such as accumulation of homocysteine, Increased concentration of 5-methyltetrahydrofolate, decrease in urinary methionine and S-adenosyl-L-methionine (SAM) concentrations, overproduction of epileptogenic and excitotoxic quinolinic acid, large reductions in melatonin synthesis, etc., that should be mentioned in the Introduction. These metabolic abnormalities could underlie several comorbidities frequently associated to ASD, such as seizures, sleep disorders, and gastrointestinal symptoms, and could contribute to autism severity.
From your study what would you propose as biomarkers in the Discussion?
Author Response
Please see the attachment for our reply to the editor and reviewer 1

Reviewer 2 Report
In the manuscript “ The autism spectrum disorder-associated bacterial metabolite 2 p-cresol derails the neuroimmune response of microglial cells 3 partially via reduction of ADAM17 and ADAM10”, Zheng et al. described the use of a neuroinflammation model of LPS-activated BV2 microglia to unveil the potential molecular mechanism of p-cresol in autism spectrum disorder pathogenesis. The aim of the study was to identify the mechanisms of the bacterial metabolite 4-methylphenol (p-cresol) derivative in autism spectrum disorder pathogenesis. The experiments appear generally well conducted. Some questions need to be addressed before the acceptance of the paper, including some data analysis.
I have the following comments that should be addressed:
Major points
The study is very interesting and relevant to the field, however, it needs to be rewritten for clarification, there are many errors that showed a lack of attention.
In the introduction, the authors say, “there is little known about the involvement of ADAM10 in ASD” but they do not show any example. We know that ADAM10 plays a key role in the modulation of the molecular mechanisms responsible for dendritic spine formation, maturation and stabilization and in the regulation of the molecular organization of the glutamatergic synapse. Therefore, it is not a surprise that an alteration of ADAM10 activity correlated to different types of synaptopathies, including some neurodevelopmental disorders such as ASD. The author could explain better the role of ADAM10 during the development and in ASD.
Line 102: The authors says, “ADAM10 and ADAM10 is affected in…” I think the authors meat ADAM10 and ADAM17.
The material and Methods session needs to be improved.
Line 103: The phrase “pathogenesis of in utero VPA-exposed male mice, the brain 103 tissues of VPA-induced mice were used to measure...” is confusing please rewrite for clarification.
In the material and Methods session, where the authors described the BV2 microglial cell viability some aspects are not clear. For example, in line 115 the authors say, “To assess the effect of pCS on cell viability, 5,000 BV2 cells/well were incubated with pCS…”, however, the authors do not explain which plate, I assume it is 96 well plate as they mentioned it later., but it is not clear. The methodology can be improved to help others to reproduce their findings.
In the session BV2 microglial ADAM10 and ADAM17 expression and inflammatory response the authors say, “IL-6, 5,000 BV2 cells/well were seeded in a 96-well plate”. In the next session Phagocytosis activity assay, the authors say, “50,000 BV2 cells/well were seeded into 96-well plate”. Have the authors used 5,000 BV2 cells/well or 50,000 in a 96-well plate?
In Figure 2 the authors showed the effect of ADAM10 inhibitor (GI254023X) or ADAM17 inhibitor (TAPI-1) on the release 280 of TNF-α, TNFR-1 and IL-6R by constitutively and under inflammation BV2 microglia. The authors showed that the TAPI-1 increases the IL6 in comparison with LPS. How do you explains it?
The discussion needs to be improved.
Minor mistakes:
In line 121 the authors wrote 200uL DMSO instead of 200 µL.
The graphs are too small it is difficult to see them; you could increase the size of all the graphs.
Author Response
Please see the attachment for our reply to the editor and reviewer 2

Round 2
Reviewer 1 Report
Authors heve changed what was required. The paper can be now be publisched